# Evaluation of the Antidiabetic Potential of Xanthone-Rich Extracts from *Gentiana dinarica* and *Gentiana utriculosa*

**DOI:** 10.3390/ijms25169066

**Published:** 2024-08-21

**Authors:** Jelena Arambašić Jovanović, Dijana Krstić-Milošević, Branka Vinterhalter, Svetlana Dinić, Nevena Grdović, Aleksandra Uskoković, Jovana Rajić, Marija Đorđević, Ana Sarić, Melita Vidaković, Mirjana Mihailović

**Affiliations:** 1Depatment of Molecular Biology, Institute for Biological Research “Siniša Stanković”, National Institute of Republic of Serbia, University of Belgrade, Bulevar Despota Stefana 142, 10060 Belgrade, Serbia; sdinic@ibiss.bg.ac.rs (S.D.); nevenag@ibiss.bg.ac.rs (N.G.); auskokovic@ibiss.bg.ac.rs (A.U.); marija.sinadinovic@ibiss.bg.ac.rs (M.Đ.); ana.saric@ibiss.bg.ac.rs (A.S.); melita@ibiss.bg.ac.rs (M.V.); mista@ibiss.bg.ac.rs (M.M.); 2Department of Plant Physiology, Institute for Biological Research “Siniša Stanković”, National Institute of Republic of Serbia, University of Belgrade, Bulevar Despota Stefana 142, 10060 Belgrade, Serbia; horvat@ibiss.bg.ac.rs

**Keywords:** *Gentiana dinarica*, *Gentiana utriculosa*, xanthones, diabetes, antidiabetic properties, norswertianin, norswertianin-1-*O*-primeveroside, decussatin, decussatin-1-*O*-primeveroside

## Abstract

Despite the existence of various therapeutic approaches, diabetes mellitus and its complications have been an increasing burden of mortality and disability globally. Hence, it is necessary to evaluate the efficacy and safety of medicinal plants to support existing drugs in treating diabetes. Xanthones, the main secondary metabolites found in *Gentiana dinarica* and *Gentiana utriculosa*, display various biological activities. In in vitro cultured and particularly in genetically transformed *G. dinarica* and *G. utriculosa* roots, there is a higher content of xanthones. The aim of this study was to investigate and compare antidiabetic properties of secondary metabolites (extracts) prepared from these two *Gentiana* species, cultured in vitro and genetically transformed with those collected from nature. We compare HPLC secondary metabolite profiles and the content of the main extract compounds of *G. dinarica* and *G. utriculosa* methanol extracts with their ability to scavenge DPPH free radicals and inhibit intestinal α-glucosidase in vitro. Anti-hyperglycemic activity of selected extracts was tested further in vivo on glucose-loaded Wistar rats. Our findings reveal that the most prominent radical scavenging potential and potential to control the rise in glucose level, detected in xanthone-rich extracts, were in direct correlation with an accumulation of xanthones norswertianin and norswertianin-1-*O*-primeveroside in *G. dinarica* and decussatin and decussatin-1-*O*-primeveroside in *G. utriculosa*.

## 1. Introduction

Plants, as an inexhaustible source of natural products with various medical properties, are continuously explored for the development of novel drugs [1,2]. However, it is estimated that only a small percentage of all plant species (around 15%) have been explored for the presence of biologically active compounds with pharmacological potential [3]. These compounds are products of plant secondary metabolism and are identified as secondary metabolites or phytochemicals. Plants produce a high diversity of secondary metabolites that provide protection against herbivores, different pathogens, and abiotic stress and enable the adaptation of plants to changing environmental conditions [4,5]. Unlike primary metabolites synthesized in all plant species, secondary metabolites are specific to plant species and to plant organs, including leaves, stems, roots, and flowers [6]. Secondary plant metabolites are often classified according to their chemical structures into three major classes: phenolics, terpenes, and alkaloids [7].

The most common challenge faced when using wild plants as a source for the isolation and characterization of biologically active compounds is the low availability of the plant material from nature. Secondary metabolites are accumulated by plants in small quantities. Although sufficient for initial pharmacological evaluation, those quantities are insufficient for testing on a broad range of biological activities and for mass production. Additionally, many plant species have become endangered as a result of uncontrolled harvesting and/or extreme environmental changes. In order to obtain a sufficient amount of secondary metabolites, the most commonly used approach includes biotechnology-based in vitro techniques for plant cultivation in controlled environmental conditions, such as callus culture, hairy root culture, cell suspension culture, and micropropagation [7].

The species comprising the Gentianaceae are of great importance to the pharmaceutical industry because their phytochemical constituents display vast and versatile pharmacological effects [8]. The main secondary metabolites of Gentianaceae species are phenols (xanthones and *C*-glucoflavones) and terpenoids. Aside from Gentianaceae species, the majority of xanthones are found in only one plant family—Guttiferae [9]. Xanthones are heterotricyclic compounds substituted with simple isoprene, methoxy, and hydroxyl groups at various locations on aromatic A and B rings. In the Gentianaceae, xanthones occur as simple oxygenated xanthones (mono-, tri-, tetra-, and hexaoxygenated) and xanthone glycosides (*C*- and *O*-glycosides). Xanthone-*O*-glycosides are very frequent among the Gentianaceae species while xanthone-*C*-glycosides are quite rare [10]. Bioproduction of xanthones is of great importance since these natural compounds exhibit a variety of pharmacological and health benefits [11]. Previous research was initiated with the aim to establish protocols for shoot micropropagation, excised root cultures, and hairy root cultures of *Gentiana dinarica Beck*. and *Gentiana utriculosa* L. in order to enhance the production of secondary metabolites, especially xanthone compounds [12,13,14,15,16,17]. Compared to plants grown in nature, in vitro-cultured *G. dinarica* and *G. utriculosa* had a higher content of xanthones, particularly in genetically transformed roots (hairy roots) [12,17,18]. This is highly important for *G. dinarica* because this perennial plant species is rare and endangered, limited to the Dinaric mountains of the Balkan peninsula and the Apennines mountains in Italy [19]. On the other hand, *Gentiana utriculosa* L. is an annual plant species with a wide distribution in the mountains of central Serbia [20] and in Central Europe [21]. These two Gentians are rich in tetraoxygenated xanthones with a 1, 3, 7, 8-oxygenation pattern. Norswertianin and its *O*-glycoside (norswertianin-1-*O*-primeveroside) are typical xanthones for *G. dinarica* [6,12], while decussatin (1-hydroxy-3,7,8-trimethoxyxanthone) and its *O*-glycoside (decussatin-1-*O*-primeveroside) are characteristic for *G. utriculosa* [22]. These xanthones display at least one of the various biological activities, e.g., antimicrobial [23], radioprotective [24], antitumor [25], antidepressant [26], vasodilator [27], antiulcer [28], antibacterial, antifungal, antioxidative, and hypoglycemic effects [29], as well as potential to act as a chemopreventive [30] and anticancerogenic agent [31].

The curative value of plant extracts depends on the composition and different combinations of biologically active compounds [32]. The aim of this study was to compare the antidiabetic properties of extracts derived from aerial parts and roots of *G. dinarica* and *G. utriculosa* that were collected from nature, in vitro propagated, or genetically transformed in order to find the optimal combination of active compounds for the treatment of diabetes. The importance of finding new antidiabetogenic extracts is reflected in the fact that diabetes is one of the top 10 causes of death globally and, together with cardiovascular disease, cancer, and respiratory disease, account for over 80% of all premature noncommunicable disease deaths [33]. Diabetic patients are, due to the of lack of insulin secretion and/or action, constantly exposed to high blood sugar levels, which causes oxidative stress, further leading to the development and progression of diabetes and its complications [34,35]. Because of the fact that most of the currently available antidiabetic drugs have their limitations, adverse effects, and secondary failures, the focus has been shifted towards the medical plants and their role in the therapy of diabetes.

## 2. Results

### 2.1. Qualitative and Quantitative Analysis of Secondary Metabolites in G. dinarica (Gd) and G. utriculosa (Gu) Extracts

The secondary metabolite profiles of Gd and Gu methanol extracts analyzed using the HPLC-DAD method are presented in Figure 1, and the chemical structures of compounds identified in these extracts are presented in Figure 2. It is evident that chromatograms of *G. dinarica* extracts derived from vegetative roots (Gd4) and from genetically transformed roots (Gd5 and Gd6) (Figure 1b) contain different sets of secondary metabolites from the other three examined extracts (Gd1—aerial parts of wild growing plants; Gd2—roots of wild growing plants; Gd3—shoot culture). An analysis revealed the presence of xanthones norswertianin-1-*O*-primeveroside (6), norswertianin-8-*O*-primeveroside (7), gentioside (8), and norswertianin (10) in all examined Gd methanol extracts (Figure 1a,b). Apart from these xanthones, Gd1, Gd2, and Gd3 also contained groups of bitter glycosides, including swertiamarin (1), gentiopicrin (3), sweroside (4), and amarogentin (9) (Figure 1a). Additionally, the presence of flavones isoorientin (5) and isoorientin-4-O-glucoside (2) was detected only in these extracts (Figure 1a).

On the other hand, -1-*O* primeverosides of decussatin (13) and gentiakochianin (12) along with their corresponding aglycones decussatin (16) and gentiakochianin (15) are the main xanthones identified in Gu methanol extracts (Gu1—aerial parts of wild growing plants; Gu2—shoot culture; Gu3—genetically transformed shoots; Gu4—genetically transformed roots; Figure 1c). Xanthone *C*-glucoside mangiferin (11) was detected in all examined extracts of Gu except in Gu4 (Figure 1c).

The contents of the main secondary metabolites of Gd and Gu methanol extracts are shown in Table 1 and Table 2. The most abundant compounds in the aerial parts and shoots of *G. dinarica* (Gd1 and Gd3) were secoiridoids—swertiamarin, gentiopicrin, and sweroside. The amount of gentiopicrin was the most dominant. Xanthones (norswertianin, norswertianin-1-*O*-primeveroside, and gentioside) were the second group of secondary metabolites present in these extracts with much lower content (Table 1). The preponderant compound in extracts obtained from vegetative roots (Gd4) and transformed roots (Gd5 and Gd6) was norswertianin-1-*O*-primeveroside, followed by its aglycone norswertianin. However, the predominant compound in the extract of roots from wild plants was gentiopicrin, which had double the amount of norswertianin-1-O-primeveroside. The rest of the secondary metabolites (swertiamarin, sweroside, gentioside, and norswertianin) were quantified in much lower content (Table 1).

Quantitative analysis of secondary metabolites in *G. utriculosa* extracts showed high amounts of xanthones decussatin-1-O-primeveroside and its corresponding aglycone decussatin in shoot culture (Gu2) and genetically transformed shoots (Gu3) and roots (Gu4) (Table 2). The highest amount of decussatin-1-O-primeveroside was recorded in the extract of genetically transformed roots (Gu4). Xanthone mangiferin was detected in higher content in Gu2 extract (Table 2).

### 2.2. DPPH Radical Scavenging

The in vitro antioxidant activity of the extracts was determined using a DPPH assay. As can be seen from Figure 3, extracts from *G. dinarica* vegetative and transgenic roots (Gd4, Gd5, and Gd6) displayed significantly higher scavenging activity than the other three extracts of *G. dinarica* (Gd1, Gd2, and Gd3) and all four extracts of *G. utriculosa*. Although DPPH radical scavenging activity increased with increasing concentrations of all examined extracts (Figure 3), only vegetative and transformed root extracts of *G. dinarica* displayed the DPPH radical scavenging activity at the same concentration as ascorbic acid, which served as the reference substance. These three Gd extracts reached maximal scavenging (90%) at the concentration of 1 mg/mL, while ascorbic acid reached its maximum (92%) at four times lower concentration. The remaining three extracts of *G. dinarica* and all four extracts of *G. utriculosa* showed scavenging of DPPH radicals at ten times higher concentrations than the reference substance. Among these extracts, the best activity (91%) was detected in the aerial parts of wild grown plants (Gd1) and the lowest activity (62%) in the shoot culture of *G. dinarica* (Gd3). In contrast, all examined extracts of *G. utriculosa* showed similar scavenging activity (Figure 3).

The compounds responsible for free radical scavenging activity were determined using the HPLC method with pre-chromatographic reaction of tested extracts and DPPH radicals. Figs. 4 and 5 show the comparative chromatograms of *G. dinarica* and *G. utriculosa* extracts before and after the reaction with DPPH radicals. The decrease in peak areas in the chromatogram profile of the extract revealed radical scavenging compounds. These compounds are oxidized in reaction with DPPH, which causes a decrease in their concentration. A further consequence of that reaction is a decrease in peak areas. In the *G. dinarica* extract, the largest decrease was recorded for the peak area of norswertianin-1-O-primeveroside (1) and its aglycon norswertianin (4), indicating that these xanthones had the highest antioxidant activity. The strong antioxidant activity of Gd extract is obviously correlated with the high content of these xanthones (Figure 4). However, in *G. utriculosa* extract, the peak area of xanthones was not significantly reduced, pointing out that these compounds had moderate antioxidant activity (Figure 5).

### 2.3. The Inhibitory Effect of Gd and Gu Extracts on α-Glucosidase

The results in Figure 6 demonstrate the percentage inhibition of six *G. dinarica* extracts and four *G. utriculosa* extracts against the intestinal α-glucosidase (maltase). At a concentration of 1 mg/mL, examined extracts had moderate inhibitory potential towards α-glucosidase, ranging from 4.5 to 11.33% for Gd extracts and 4.56 to 28.29% for Gu extracts. On the other hand, at a concentration of 5 mg/mL, the examined extracts markedly inhibited intestinal maltase, ranging from 32.43 to 89.59% for Gd extracts and 9.36 to 64.4% for Gu extracts. The intestinal maltase inhibitory activity of Gd extracts was in the following order, from highest to lowest: Gd4, Gd5, Gd6 > Gd2 > Gd1, Gd3; for Gu extracts: Gu4 > Gu2, Gu3 > Gu1.

### 2.4. Antihyperglycemic Activity of the Gd and Gu Extracts on Oral Glucose-Loaded Rats

Further analysis of the antihyperglycemic activity of the Gd and Gu extracts was conducted on selected extracts (Gu2, Gu3, Gd4, Gd5) based on their success in DPPH radical scavenging and in inhibition of alpha-glucosidase enzyme. In addition, the chosen extracts from both plants had to meet the criteria of coming from the same part of the plant but being grown in two different ways (Gu2, Gd4—in vitro propagation and Gu3, Gd—genetic transformation). The results of the oral glucose tolerance test in normal Wistar rats are shown in Figure 7 and Figure 8. As seen in Figure 7, the initial blood glucose levels of all groups prior to drug/extract administration were equal. Following oral glucose loading (2 g/kg) to control and test groups, the blood glucose level was elevated to the maximum level after 30 min in vehicle-treated rats and decreased subsequently over time. At 15 min post-oral glucose loading, only the standard drug (glibenclamide (GLC), 5 mg/kg) treated group, and groups treated with Gu2 (400 mg/kg) and Gu3 (400 mg/kg) extracts showed a significant decline in blood glucose levels compared to the vehicle group. Likewise, the GLC 5 mg/kg-treated group and all groups treated with extracts in a dose of 400 mg/kg (Gu2, Gu3, Gd4, Gd5) showed a significant antihyperglycemic effect 30 min after glucose loading compared to the vehicle-treated group (Figure 7). In extract-treated groups, the blood glucose reached a maximum level at 60 min post-oral glucose loading. In addition, the area under the curve (AUC) in the first 60 min was significantly reduced in GLC and all extract-treated groups when compared to the vehicle group (Figure 8).

## 3. Discussion

Diabetes mellitus is one of the major multifactorial health problems globally and is associated with high morbidity and mortality rates. Insufficient blood sugar regulation over time leads to serious damage to the heart, blood vessels, eyes, kidneys, and nerves, causing chronic complications in diabetic patients. Conventional antidiabetic drugs improve blood glucose levels and the survival of people with diabetes but do not prevent secondary chronic complications associated with diabetes leading to an increase in the number of people living with more than one chronic condition, known as multiple long-term conditions (MLTC) [36]. Also, the side effects of conservative therapy impose restrictions on the choice of an antidiabetic drug. Therefore, despite the effectiveness of different antihyperglycemic drugs, new antidiabetogenic agents are increasingly sought among medicinal plants. In this study, the analysis of the antidiabetic potential of two *Gentiana* plant species, *G. dinarica* and *G. utriculosa*, and the association with secondary metabolite composition of their methanolic extracts revealed that xanthone-rich extracts have the most prominent antioxidant and antihyperglycemic effects.

Based on the quantitative and qualitative analysis of the extracts obtained from these two *Gentiana* species, it can be noticed that the accumulation of secondary metabolites is species-specific. Also, the quantitative and qualitative composition changes depending on the part of the plant the secondary metabolites were isolated from (aerial part or root) and the way a certain part of the plant was obtained (from nature, in vitro propagation, or genetic transformation). Differences in secondary metabolite composition between different *Gentiana* species and between underground and aerial parts of the same plant are already documented in the literature [37,38]. In accordance with previous studies [12], the dominant compounds of the extracts are derived from *G. dinarica* are secoiridoids (sweroside, swertiamarin, and gentiopicroside) and/or xanthones (norswertianin, norswertianin-1-O-primeveroside, and gentioside). Correspondingly to earlier research, in vitro culture conditions significantly increased the accumulation of all quantified secondary metabolites, especially gentiopicrin in shoot cultures compared to samples of plants collected in nature [37,39,40,41,42]. On the other hand, as was reported earlier, in vitro propagation and genetic modification by Agrobacterium rhizogenes stimulated the production of norswertianin and norswertianin-1-O-primeveroside in transgenic roots of *G. dinarica* [43] and inhibited the production of secoiridoids and gentioside. Although genciopicrin is the main bitter constituent of Gentian roots [44], it was not detected either in vegetative or transgenic roots. In comparison to *G. dinarica*, *G. utriculosa* accumulates completely different sets of secondary metabolites, among which the most abundant were decussatin, decussatin-1-O-primeveroside, and mangiferin. This is in accordance with previous reports [17,22]. Although, according to Jensen and Schripsema [9], at least one bitter compound from the class of iridoids should be found in practically all *Gentiana* species, we did not detect any in *G. utriculosa*.

Hyperglycemia and glucose autooxidation in diabetes are responsible for the overproduction of various reactive oxygen species, causing oxidative damage to macromolecules such as lipids, proteins, and DNA. The inability of the body to effectively remove free radicals by antioxidant processes leads to oxidative stress, which affects the majority of tissues and organs in diabetic patients and promotes diabetic complications [35]. Given this, the use of natural products with antioxidant properties could have multiple beneficial effects on diabetic patients. The results of the antioxidant activity of the extracts from *G. dinarica* and *G. utriculosa*, measured through a DPPH scavenging assay, showed that extracts of both plants can scavenge the radical to a certain extent. The greatest scavenging activity, recorded in roots of *G. dinarica* obtained in vitro and by genetic transformation, was attributed to xanthones norswertianin and norswertianin-1-O-primeveroside. Uvarani et al. [45] also showed that norswertianin isolated from Swertia corymbosa (Gentianaceae) has the ability to scavenge ROS (DPPH, OH-, and NO-) and they ascribed this strong antioxidant capacity of norswertianin to the presence of the catechol moiety, which enhanced its H-donating ability. The other three *G. dinarica* extracts showed approximately ten times weaker ability to scavenge free radicals, which was in accordance with their very low level of norswertianin and norswertianin-1-O-primeveroside. Four extracts obtained from *G. utriculosa* also showed ten times weaker ability to scavenge DPPH radicals in comparison to extracts obtained from roots of *G. dinarica* (vegetative root culture and transgenic roots). Shoot culture and transgenic shoots showed the strongest scavenging activity among *G. utriculosa* extracts and this scavenging activity was attributed mainly to xanthones mangiferin, decussatin, and decussatin-1-O-primeveroside. Mangiferin is one of the most studied xanthone-C-glycosides with several beneficial properties, including antioxidant activity [46]. Unlike mangiferin, whose DPPH scavenging activity has been shown previously [47], there are no data published about radical scavenging activity and antioxidant properties of decussatin and decussatin-1-O-primeveroside.

Another effective strategy for diabetes management is the inhibition of intestinal α-glucosidase enzyme, which slows down the digestion and absorption of complex carbohydrates and in turn alleviates the increase in postprandial glycemia. Commercial drugs currently used as reversible inhibitors for α-glucosidase inhibition, such as acarbose, miglitol, and voglibose, exhibit side effects such as abdominal distension, bloating, flatulence, and possibly diarrhea [48]. Therefore, there has been a growing interest in research seeking novel new plant-derived inhibitors with improved efficacy [49]. In this study, both investigated *Gentiana* species possessed antidiabetic properties regarding the inhibition of the α-glucosidase enzyme, with different extracts having different inhibitory abilities. All extracts showed dose/concentration-dependent effects of inhibition on the activity of the α-glucosidase enzyme, suggesting a competitive type of inhibition [50]. The highest inhibitory activity recorded in vegetative and transgenic roots of *G. dinarica* was attributed to xanthones norswertianin and, particularly, norswertianin-1-*O*-primeveroside. In a previous investigation by Uvarani et al. [45], it was shown that norswertianin exhibited the most prominent inhibition of the α-glucosidase enzyme among six other xanthones isolated from *Swertia corymbosa*. The same group of authors ascribed this inhibition to the presence of four free OH groups involved in H-bonding interactions with the α-glucosidase enzyme [45], which can be a valid explanation also for norswertianin-1-*O*-primeveroside having three free OH groups. Also, the root extract obtained from the wild growing plant showed a significant inhibitory activity that can be associated with a substantial amount of norswertianin-1-*O*-primeveroside. Among four extracts derived from *G. utriculosa*, which were less effective than the *G. dinarica* root extract, the highest inhibition of α-glucosidase enzyme was recorded in the extracts from genetically transformed roots that were rich in decussatin and decussatin-1-*O*-primeveroside. Although decussatin was found to have hypolipidemic and hypoglycemic effects in male Wistar rats fed with a high-fructose diet, the reported mechanism of action has not been through the inhibition of the α-glucosidase enzyme [51]. The other three *G. utriculosa* extracts were less effective in the inhibition of the α-glucosidase enzyme due to lower levels of decussatin and, particularly, decussatin-1-*O*-primeveroside. On the other hand, these extracts had mangiferin, a compound known for its ability to inhibit the α-glucosidase enzyme [52].

Sustained reduction in hyperglycemia is the key factor for preventing or reversing micro- and macrovascular complications, thus improving the quality of life in diabetic patients [53]. Therefore, we selected the glucose-induced hyperglycemic model to screen the antihyperglycemic activity of the Gd and Gu extracts. To determine the ability of the selected extract of these two *Gentiana* species to reduce the increased blood glucose levels, oral glucose tolerance tests were performed [54]. The criterion for the selection of the extracts to be used in further analysis was the success in DPPH radical scavenging and in the inhibition of the alpha-glucosidase enzyme. This study revealed that treatment with all four extracts at a dose level of 400 mg/kg had significantly lowered blood glucose levels in the first 60 min after glucose loading compared to the vehicle control. Treatment with extracts delayed and lowered the postprandial glucose spike, which indicates that extract-treated rats had increased glucose utilization, lower glucose absorption, or decreased glucose production from the liver compared to the control animals [55,56]. The mechanism behind this antihyperglycemic activity of *G. dinarica* and *G. utriculosa* extracts involves an insulin-like effect [51] or prevention of glucose transport at the level of some transport protein on the absorptive intestinal surface. It is known that the absence of the SGLT1 transporter prevents the absorption of glucose at the level of the small intestine in humans [57]. The main secondary metabolites in the examined extracts of these two Gentians are norswertianin and norswertianin-1-*O*-primeveroside for Gd extracts and mangiferin, decussatin, and decussatin-1-*O*-primeveroside for Gu extracts. Some of these compounds are known from the literature for their hypoglycemic effects in vivo. First of all, it has been shown that mangiferin facilitates β-cell proliferation and islet regeneration in mice with 70% partial pancreatectomy [58]. Decussatin from *Lomatogonium rotatum* (Gentianaceae) was found to produce hypolipidemic and hypoglycemic effects in male Wistar rats fed with a high fructose diet (51) and this effect has been attributed to its potential to modulate AMP-activated protein kinase (AMPK) activity in the liver. To date, no in vivo studies have been performed regarding the antihyperglicemic effects of norwertianin and norswertianin-1-*O*-primeveroside.

## 4. Materials and Methods

### 4.1. Plant Material

Plant extracts used in this study were obtained from the plant material of two *Gentiana* species collected in nature and cultivated in vitro. Extracts Gd1 (aerial parts of wild growing plants) and Gd2 (roots of wild growing plants) were derived from plants of *Gentiana dinarica* (Gd) collected in June 2015 on Mount Tara (~1300 m), western Serbia. A voucher specimen (accession number Gd072001) was deposited in the herbarium at the Faculty of Biology, University of Belgrade, Code BEOU. Extract Gd3 was obtained from *G. dinarica* shoot culture [13]. Extract Gd4 was derived from *G. dinarica* vegetative root cultures, while extracts Gd5 and Gd6 were prepared from genetically transformed roots—clone B and clone 3, respectively. Genetic transformation of *G. dinarica* and establishment of vegetative and transgenic root cultures were previously published by Vinterhalter et al. 2015 [14].

Extract Gu1 (aerial parts of wild growing plants) was derived from *Gentiana utriculosa* collected on Divčibare Mountain (at ca. 800 m) in Serbia in June 2016. A voucher specimen (accession number Gu072002) was deposited in the herbarium at the Faculty of Biology, University of Belgrade, Code BEOU. Extract Gu2 was prepared from *G. utriculosa* shoot cultures [16], while extracts Gu3 and Gu4 were derived from genetically transformed shoots and transgenic roots, respectively. The protocol for the establishment of transgenic roots by *Agrobacterium rhizogene*-mediated transformation was reported by Vinterhalter et al. 2019 [17].

### 4.2. Extraction and HPLC Analysis of Secondary Metabolites

Secondary metabolite identification and content determination were performed in extracts from in vitro-derived plant material and wild growing plants. Plant material was air-dried at room temperature, ground using an electric mill, and extracted with methanol in an ultrasonic bath for 20 min. After sonication, extraction was continued by maceration in the dark at room temperature for 24 h. Then, extracts were filtered and evaporated to dryness in vacuum rotary evaporator (Buchi R-210, Flawil, Switzerland) at 50 °C. The extracts were stored at 4 °C until further analysis.

Identification and quantification of secondary metabolites of *G. dinarica* and *G. utriculosa* extracts were accomplished with chromatographic analysis (Agilent series 1100 HPLC instrument with a diode array detector, Waldbronn, Germany) on a reverse phase Zorbax SB-C18 (Agilent) analytical column (150 mm × 4.6 mm i.d., 5 μm particle size) thermostated at 25 °C. The mobile phase consisted of solvent A (0.1%, *v*/*v* solution of orthophosphoric acid in water) and solvent B (acetonitrile, J.T. Baker, Deventer, The Netherlands).

The separation of the components was performed using an elution gradient according to the earlier published method by Krstic-Milosevic et al. [59]: 98–90% A 0–5 min, 90% A 5–10 min, 90–85% A 10–13 min, 85% A 13–15 min, 85–70% A 15–20 min, 70–40% A 20–24 min, 40–0% A 24–28 min. The injection volume was 5 μL. Detection wavelengths were set at 260 and 320 nm, and the flow rate was 1 mL min^−1^. The isolation and characterization of xanthones norswertianin-1-*O*-primeveroside and gentioside from *G. dinarica* roots were previously reported [60]. Acid hydrolysis of norswertianin-1-*O*-primeveroside with 2N HCl yielded xanthon aglycone norswertianin. Standards of xanthones decussatin-1-*O*-primeveroside and decussatin were previously isolated from the aerial parts of *G. utriculosa* (22). Xanthone mangiferin was purchased from Sigma-Aldrich, Steinheim, Germany. Commercial standards of secoiridoids swertiamarin, gentiopicrin, and sweroside were bought from Cfm Oscar Tropitzsch (Bayern, Germany). Identification of secoiridoids and xanthones was confirmed using a co-injection method using standard compounds. Quantification of secondary metabolites was performed using calibration curves in an external standard method. Standard solutions for HPLC were prepared by dissolving standard compounds in methanol. All experiments were repeated at least two times. The results are presented as mg per g of dry weight of extracts.

### 4.3. 2,2-Diphenyl-1-Picrylhydrazyl (DPPH) Free Radical-Scavenging Assay

The free radical scavenging activity of the *G. dinarica* and *G. utriculosa* methanol extracts was determined according to a standard method of Blois [61] by measuring the decrease in the absorbance of stable free radical DPPH (2,2-Diphenyl-1-picrylhydrazyl) at 517 nm as previously performed [62]. Stock solution of extracts Gd1, Gd2, Gd3, Gu1, Gu2, Gu3, and Gu4 were diluted to a concentration of 0.625, 1.25, 2.5, and 5 mg/mL in methanol. Stock solution of extracts Gd4, Gd5, and Gd6 were diluted to a concentration of 0.05, 0.1, 0.25, and 0.5, 1 mg/mL in methanol. DPPH methanolic solution (0.5 mL, 0.25 mM) was added to a mixture of 0.1 mL of sample solutions of different concentrations and 0.4 mL Tris HCl (100 mM, pH 7.4) and allowed to react at room temperature. After 30 min, the absorbance values were measured at 517 nm. Ascorbic acid was used as a standard. All tests were performed in triplicate. The inhibitory percentage of DPPH was calculated according to the following formula: percentage of inhibition = ((A blanc − A test)/A blanc) × 100, where A blanc is the absorbance of the methanolic DPPH solution and A test is the absorbance of DPPH in the solution with the extract or a standard (ascorbic acid).

### 4.4. The DPPH-HPLC Procedure

The compounds of extracts that were included in free radical scavenging activity were evaluated using the HPLC method with a pre-chromatographic reaction of tested extracts with DPPH radicals, according to the method described by Olennikov et al. [63]. The reaction mixture contained 300 µL of plant extract dissolved in methanol (10 mg/mL) and 300 µL of DPPH methanol solution (2.5 mg/mL). The control mixture consisted of 300 µL of plant extract dissolved in methanol (10 mg/mL) and 300 µL of methanol instead of DPPH solution. After incubation for 20 min at room temperature in the dark, mixtures were filtered through a 0.45 µm membrane filter and analyzed by the above-described HPLC method.

### 4.5. The Assessment of Inhibitory Effect of Gd and Gu Extracts on α-Glucosidase

The assessment of intestinal α-glucosidase inhibitory activity was based on the modified method previously described [64]. Briefly, 50 mg of rat intestinal acetone powder was homogenized in 1.5 mL of 0.9% NaCl solution. The solution was centrifuged at 12,000× *g* for 30 min and then used as the small intestinal glucosidases for maltose hydrolysis. The crude enzyme solution (20 μL) was incubated with 30 μL maltose (86 mM) and 10 μL of the extract at various concentrations, followed by the addition of 0.1 M phosphate buffer with pH 6.9 to a final volume of 200 μL. The reaction was incubated at 37 °C for 30 min. Thereafter, the mixtures were suspended in boiling water for 10 min to stop the reaction. The concentrations of glucose released from the reaction mixtures were determined with the glucose oxidase method with absorbance at a wavelength of 450 nm. Intestinal α-glucosidase inhibitory activity was expressed as the percentage inhibition using the following formula: percentage of inhibition = (A control − A sample)/A control) × 100, where A control was the absorbance without a sample and the A sample was the absorbance of the sample extract or standard (acarbose).

### 4.6. Anti-Hyperglycemic Activity of Selected Gu and Gd Extracts In Vivo

Experiments were performed on 2.5-month-old male Wistar albino rats weighing 220–250 g. All animals were kept under standard laboratory conditions in a climate-controlled room with a temperature of 24–26 °C, 20–60% relative humidity, and on a 12 h–12 h light–dark cycle. They were allowed free access to a standard chow diet and fresh water. All experimental procedures were approved by the Ministry of Agriculture, Forestry and Water Management—Veterinary Administration (protocol code 323-07-00055/2023-05); date of approval 9 January 2023) based on positive opinions of the Ethical Committee for the Use of Laboratory Animals of the Institute for Biological Research “Siniša Stanković”, National Institute of the Republic of Serbia, University of Belgrade, which acts in accordance with the Directive 2010/63/EU on the protection of animals used for experimental and other scientific purposes. For the realization of this in vivo experiment, 30 male Wistar rats, aged 2.5 months, were needed. Rats were randomly assigned to the appropriate experimental groups (five animals in each group). The first group served as the glucose control group to which glucose solution (2 g/kg) was administered by oral gavage. The second group served as the standard control group to which glibenclamide (GLC, 5 mg/kg, oral gavage) was administered. The third group received Gu2 extract isolated from in vitro-propagated shoots. The fourth group received Gu3 extract isolated from genetically transformed shoots. The fifth group received Gd4 extract isolated from in vitro-propagated roots. The sixth group received Gd5 extract isolated from genetically transformed roots. All extracts were applied in a dose of 400 mg/kg by oral gavage. Animals were fasted overnight prior to the experiment and in the morning the fasting blood glucose levels were measured with a blood glucometer (Accu-Chech Active, Roche Diabetes Care, Mumbai, India). Test extracts and reference standard drugs were given to the respective groups of animals as per their body weight. After 30 min of extract/drug administration, glucose solution was administered to all groups orally by dissolving it in distilled water. Thereafter, blood glucose level was recorded at 30, 60, 90, 120, and 180 min of post-glucose overload for accessing the antihyperglycemic activity of tested extracts and standard drugs [50]. Blood samples were obtained from the tail. Less than 2 mm of tissue was cut from the tail tip, distal to the bone, with sharp scissors. Blood was obtained by direct flow or by gently massaging the tail and collecting the blood directly on a glucose test strip of blood glucometer (Accu-Chech Active, India).

### 4.7. Statistical Analysis

Experimental data subjected to statistical analysis were expressed as the means ± S.E.M. (standard error of mean). One-way analysis of variance (ANOVA) followed by Tukey’s tests were used for multiple comparison. Statistical analysis was conducted in GraphPad Prism 8 software. Different letters indicate significant differences between extracts (*p* < 0.05).

## 5. Conclusions

The findings of this present study showed that all examined extracts obtained from *G. dinarica* and *G. utriculosa* showed moderate to strong antioxidant and antihyperglycemic effects correlated to the secondary metabolites composition, which differed among different parts of the plant (aerial vs. root) and was influenced by the way in which a certain part of the plant was obtained (from nature, in vitro propagation or genetic transformation). The most prominent antidiabetic properties were detected in extracts derived from vegetative and transgenic root cultures of *G. dinarica* and shoot and transgenic shoot cultures of *G. utriculosa*. The examined antidiabetic properties are in direct correlation with accumulation of xanthones norwertianin and norswertianin-1-*O*-primeveroside in *G. dinarica* and decussatin and decussatin-1-*O*-primeveroside in *G. utriculosa* extracts. In conclusion, the ability of xanthone-rich extracts to scavenge free radicals and control the rise in glucose levels might be potentially useful in the development of novel therapeutics for the treatment of diabetes.

## Figures and Tables

**Figure 1 ijms-25-09066-f001:**
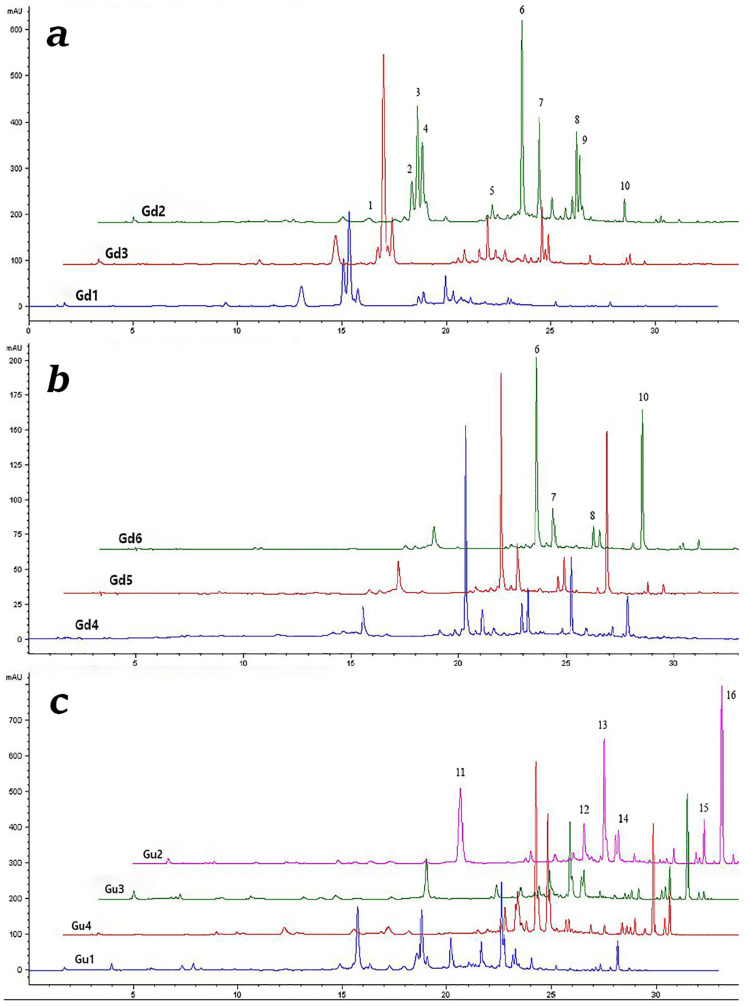
Comparative chromatograms of *G. dinarica* (Gd) (**a**,**b**) and *G. utriculosa* (Gu) (**c**) methanol extracts. (**a**) Gd1—aerial parts of wild growing plants; Gd2—roots of wild growing plants; Gd3—shoot culture. Peaks: 1—swertiamarin; 2—isoorinetin-4′-O-glucoside; 3—gentiopicrin; 4—sweroside; 5—isoorientin; 6—norswertianin-1-O-primeveroside; 7—norswertianin-8-O-primeveroside; 8—gentioside; 9—amarogentin; 10—norswertianin; (**b**) Gd4—vegetative roots; Gd5—genetically transformed roots, clone B; Gd6—genetically transformed roots, clone 3. Peaks: 6—norswertianin-1-O-primeveroside; 7—norswertianin-8-O-primeveroside; 8—gentioside; 10—norswertianin; (**c**) Gu1—aerial parts of wild growing plants; Gu2—shoot culture; Gu3—genetically transformed shoots; Gu4—genetically transformed roots. Peaks: 11—mangiferin; 12—gentiakochianin-1-O-primeveroside; 13—decussatin-1-O-primeveroside; 14—1,8-dihydroxy-3-methoxy-7-O-primeveroside; 15—gentiakochianin; 16—decussatin.

**Figure 2 ijms-25-09066-f002:**
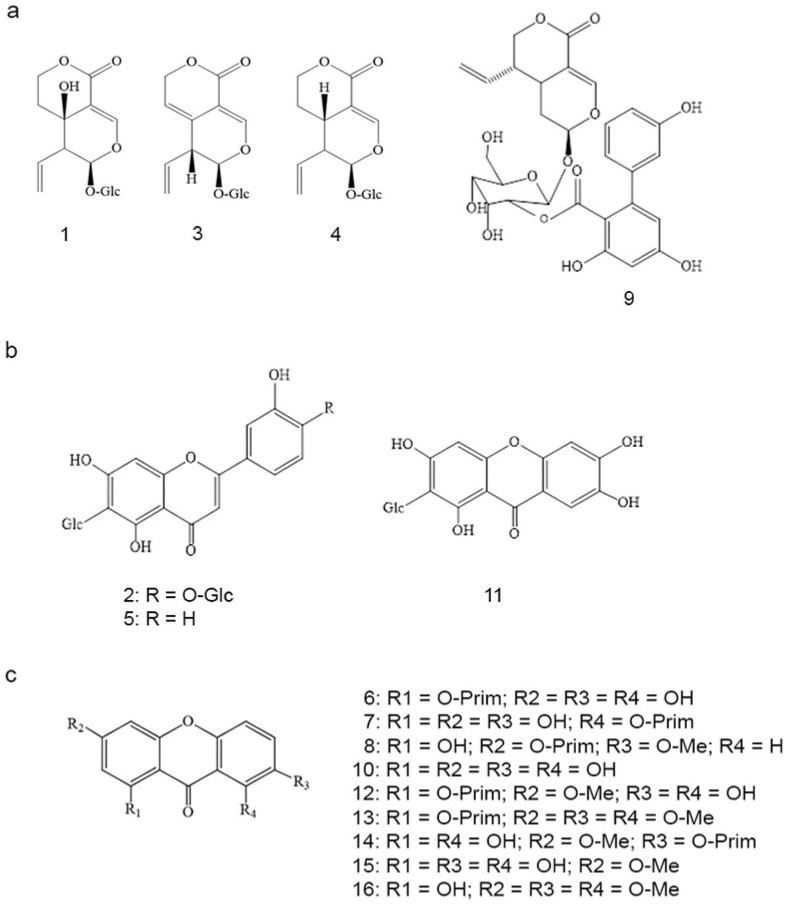
Chemical structures of compounds identified in methanol extracts of *G. dinarica* (Gd) and *G. utriculosa* (Gu). The order of compounds is in accordance with their increasing retention times in the chromatograms. (**a**) Bitter glycosides identified in *G. dinarica* methanol extracts: 1—swertiamarin; 3—gentiopicrin; 4—sweroside; 9—amarogentin; (**b**) C-glucoflavones and xanthone-C-glucoside identified in methanol extracts of *G. dinarica* and *G. utriculosa*: 2—isoorinetin-4′-O-glucoside; 5—isoorientin; 11—mangiferin; (**c**) xanthones identified in methanol extracts of *G. dinarica* and *G. utriculosa*: 6—norswertianin-1-O-primeveroside; 7—norswertianin-8-O-primeveroside; 8—gentioside; 10—norswertianin; 12—gentiakochianin-1-O-primeveroside; 13—decussatin-1-O-primeveroside; 14—1,8-dihydroxy-3-methoxy-7-O-primeveroside; 15—gentiakochianin; 16—decussatin.

**Figure 3 ijms-25-09066-f003:**
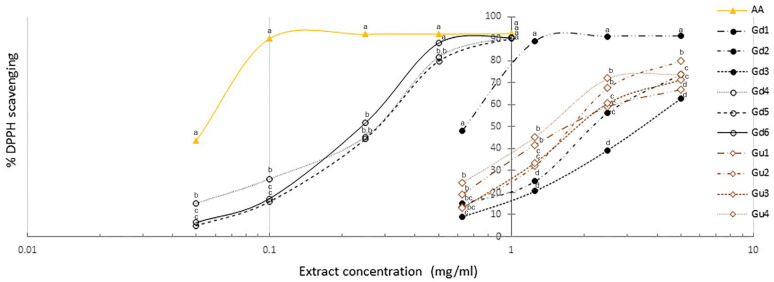
Comparative analysis of DPPH radical scavenging activity at different concentrations of methanol extracts isolated from *G. dinarica* and *G. utriculosa*. Gd1—aerial parts of wild growing plants; Gd2—roots of wild growing plants; Gd3—shoot culture; Gd4—vegetative roots; Gd5—genetically transformed roots, clone B; Gd6—genetically transformed roots, clone 3. Gu1—aerial parts of wild growing plants; Gu2—shoot culture; Gu3—genetically transformed shoots; Gu4—genetically transformed roots. The results of the assays are presented as the means ± S.E.M. from three separate measurements (n = 3). For all variables with the same superscript letter, the difference between the means (at the same concentration) is not statistically significant. If two variables have different letters, they are significantly different at *p* < 0.05.

**Figure 4 ijms-25-09066-f004:**
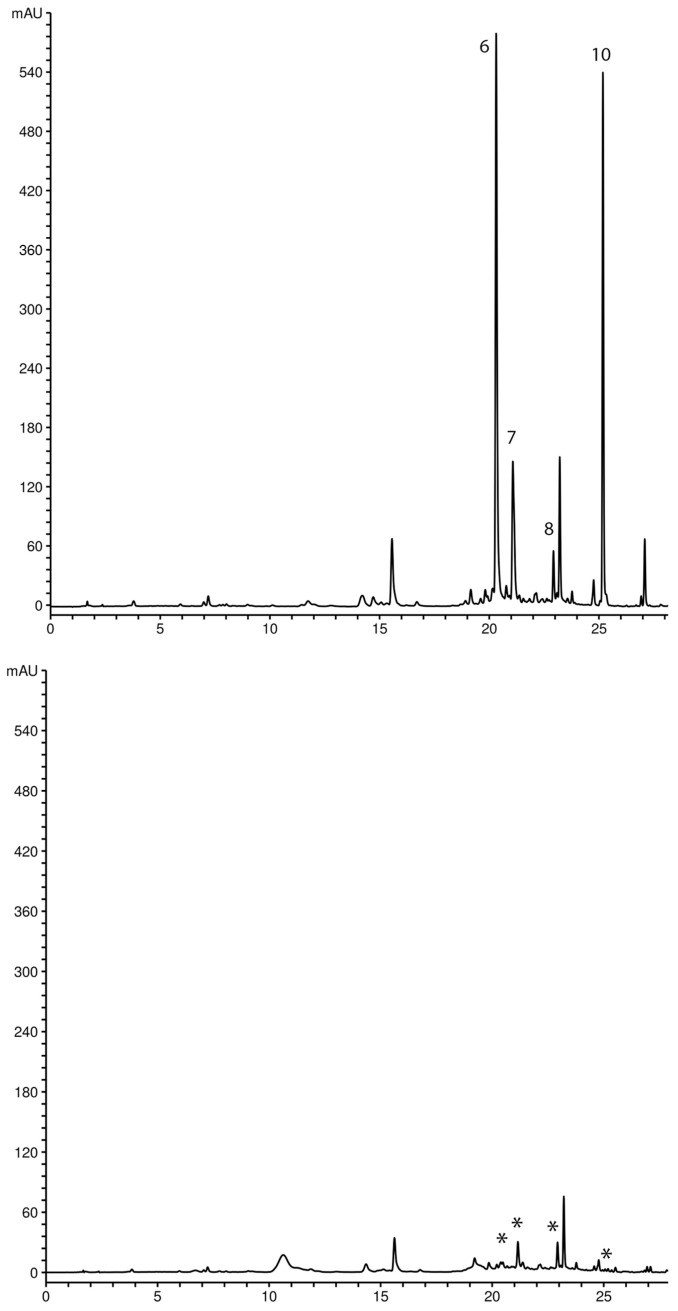
Chromatograms of *G. dinarica* extracts derived from transgenic roots clone B (Gd5) before ((**top**) chromatogram) and after reaction with DPPH radicals ((**bottom**) chromatogram). Peaks: 6—norswertianin-1-O-primeveroside; 7—norswertianin-8-O-primeveroside; 8—gentioside; 10—norswertianin. * The peaks of compounds that were involved in free radical scavenging activity.

**Figure 5 ijms-25-09066-f005:**
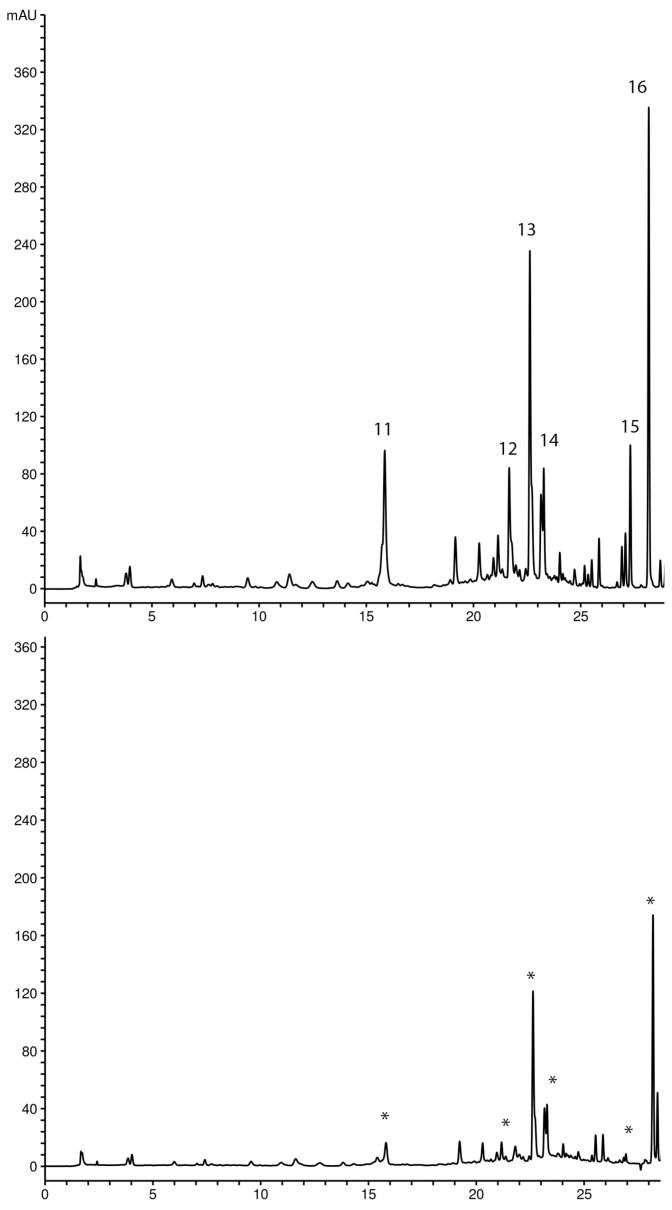
Chromatograms of *G. utriculosa* extracts derived from transgenic shoots (Gu3) before ((**top**) chromatogram) and after reaction with DPPH radicals ((**bottom**) chromatogram). Peaks: 11—mangiferin; 12—gentiakochianin-1-O-primeveroside; 13—decussatin-1-O-primeveroside; 14—1,8-dihydroxy-3-methoxy-7-O-primeveroside; 15—gentiakochianin; 16—decussatin. * The peaks of compounds that were involved in free radical scavenging activity.

**Figure 6 ijms-25-09066-f006:**
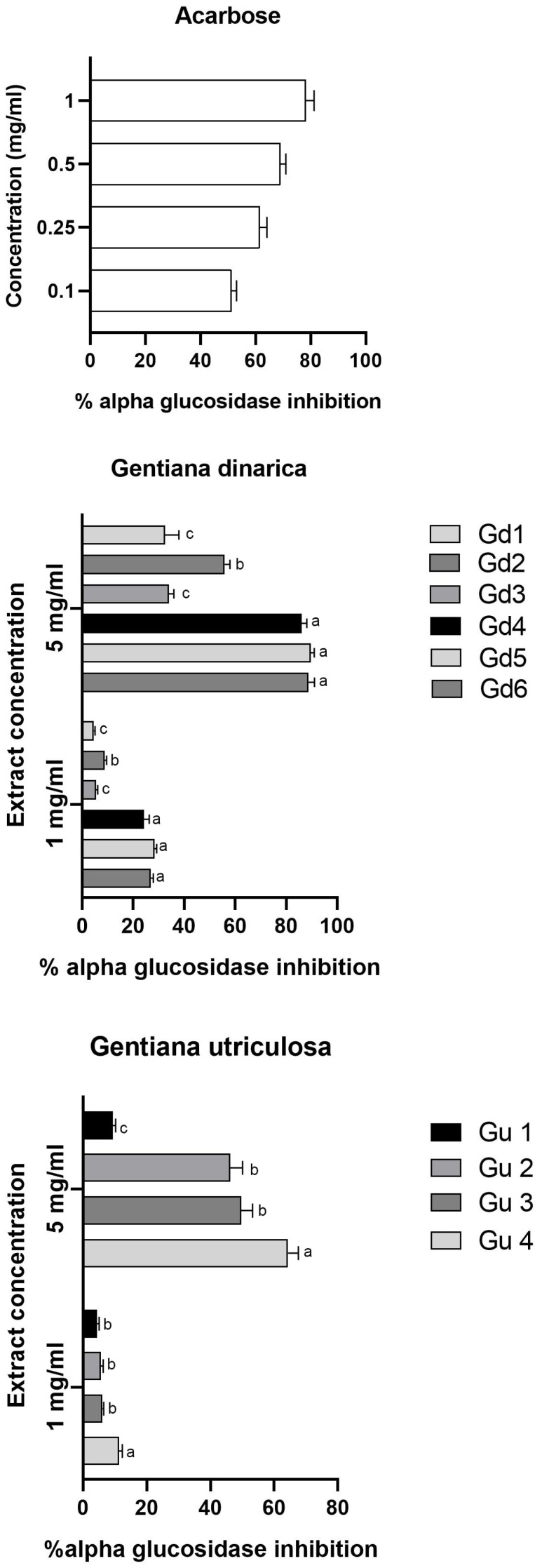
α-Glucosidase inhibitory activity at two different concentrations of methanol extracts obtained from *G. dinarica* and *G. utriculosa*. Acarbose was used as standard. Gd1—aerial parts of wild growing plants; Gd2—roots of wild growing plants; Gd3—shoot culture; Gd4—vegetative roots; Gd5—genetically transformed roots, clone B; Gd6—genetically transformed roots, clone 3. Gu1—aerial parts of wild growing plants; Gu2—shoot culture; Gu3—genetically transformed shoots; Gu4—genetically transformed roots. The results of the assays are presented as the means ± S.E.M. from three separate measurements (n = 3). For all variables with the same superscript letter, the difference between the means (at the same concentration) is not statistically significant. If two variables have different letters, they are significantly different at *p* < 0.05.

**Figure 7 ijms-25-09066-f007:**
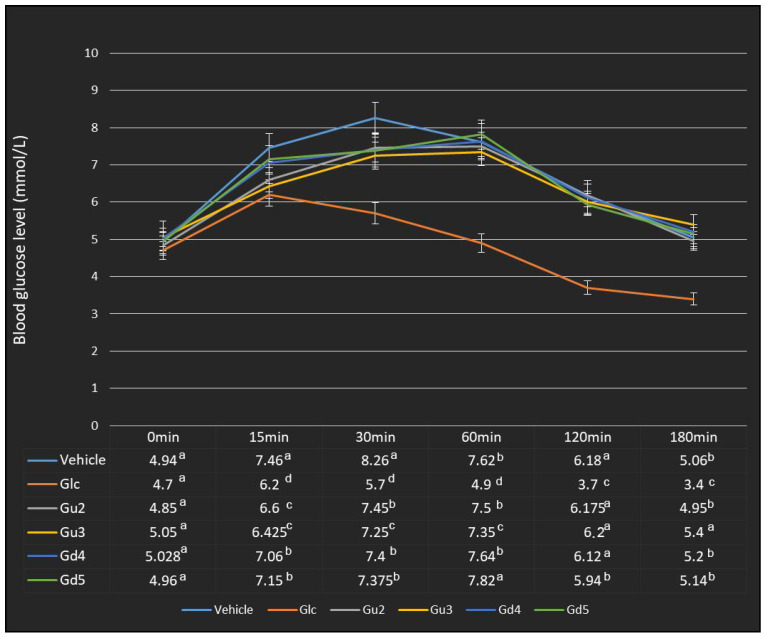
In vivo antihyperglycemic effect of selected methanol extracts of *G. dinarica* and *G. utriculosa* in oral glucose-loaded rats. Glibenclamide (Glc) was used as a control drug. Gd4—vegetative roots; Gd5—genetically transformed roots, clone B; Gu2—shoot culture; Gu3—genetically transformed shoots. Data are expressed as mean ± S.E.M.; no. of animals (N) = 5. For all variables with the same superscript letter, the difference between the means (same post-glucose overload time) is not statistically significant. If two variables have different letters, they are significantly different at *p* < 0.05.

**Figure 8 ijms-25-09066-f008:**
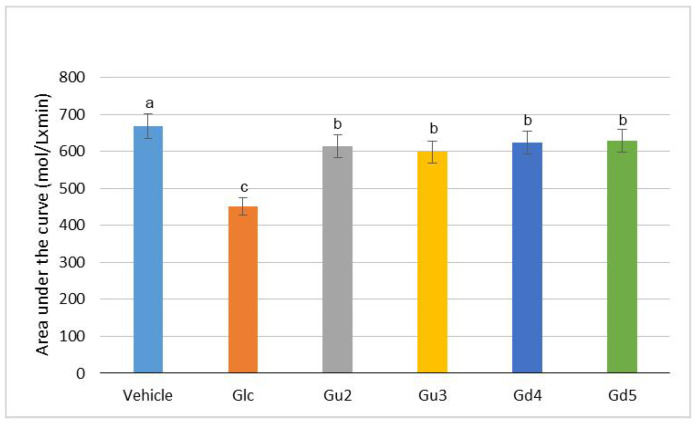
Area under the curve (AUC) for the first 60 min of oral glucose overload responses of Wistar rats treated with vehicle or selected methanol extracts of *G. dinarica* and *G. utriculosa* compared with standard drug. Glibenclamide (Glc)—standard drug. Gd4—vegetative roots; Gd5—genetically transformed roots, clone B; Gu2—shoot culture; Gu3—genetically transformed shoots. Data are expressed as mean ± S.E.M.; no. of animals (N) = 5. For all variables with the same superscript letter, the difference between the means (same post-glucose overload time) is not statistically significant. If two variables have different letters, they are significantly different at *p* < 0.05.

**Table 1 ijms-25-09066-t001:** Gd1—aerial parts of wild growing plants; Gd2—roots of wild growing plants; Gd3—shoot culture; Gd4—vegetative roots; Gd5—genetically transformed roots, clone B; Gd6—genetically transformed roots, clone 3. Values are means of three technical replicates ± S.E.M. (n = 3). For all variables with the same superscript letter, the difference between the means is not statistically significant. If two variables have different letters, they are significantly different at *p* < 0.05.

Quantitative Analysis of Secondary Metabolites in Methanol Extracts of *Gentiana dinarica*						
Extract	Swertiamarin (mg/g dw)	Gentiopicrin (mg/g dw)	Sweroside (mg/g dw)	Gentioside (mg/g dw)	Norswertianin (mg/g dw)	Norswertianin -1-O-primeveroside (mg/g dw)
Gd1	39.82 ± 1.71 ^b^	55.88 ± 2.13 ^c^	10.52 ± 0.44 ^c^	0.49 ± 0.02 ^c^	0.26 ± 0.01 ^e^	4.31 ± 0.15 ^e^
Gd2	9.91 ± 0.34 ^c^	70.15 ± 2.76 ^b^	14.46 ± 0.53 ^b^	11.37 ± 0.51 ^a^	1.95 ± 0.081 ^c^	36.65 ± 1.66 ^c^
Gd3	57.47 ± 2.44 ^a^	109.95 ± 4.17 ^a^	21.89 ± 0.98 ^a^	6.79 ± 0.29 ^b^	0.63 ± 0.02 ^d^	9.51 ± 0.35 ^d^
Gd4				0.311 ± 0.09 ^c^	19.92 ± 0.84 ^b^	126.12 ± 5.31 ^a^
Gd5					39.36 ± 1.53 ^a^	131.08 ± 4.35 ^a^
Gd6					33.78 ± 1.32 ^a^	114.86 ± 4.91 ^b^

**Table 2 ijms-25-09066-t002:** Gu1—aerial parts of wild growing plants; Gu2—shoot culture; Gu3—genetically transformed shoots; Gu4—genetically transformed roots. Values are means of three technical replicates ± S.E.M. (n = 3). For all variables with the same superscript letter, the difference between the means is not statistically significant. If two variables have different letters, they are significantly different at *p* < 0.05.

Quantitative Analysis of Secondary Metabolites in Methanol Extracts of *Gentiana utriculosa*			
Extract	Mangiferin (mg/g dw)	Decussatin (mg/g dw)	Decussatin -1-O-primeveroside (mg/g dw)
Gu1	13.12 ± 0.56 ^b^	7.44 ± 0.31 ^d^	26.64 ± 1.06 ^c^
Gu2	18.73 ± 0.84 ^a^	46.91 ± 2.15 ^b^	38.33 ± 1.64 ^b^
Gu3	7.43 ± 0.29 ^c^	27.92 ± 1.22 ^c^	23.17 ± 1.06 ^d^
Gu4		58.91 ± 2.15 ^a^	107.77 ± 3.15 ^a^

## Data Availability

The raw data supporting the conclusions of this article will be made available by the authors upon request.

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
