# Peer review of "Evaluation of the Antidiabetic Potential of Xanthone-Rich Extracts from *Gentiana dinarica* and *Gentiana utriculosa"

_ijms, 2024, doi:10.3390/ijms25169066_

Round 1

Reviewer 1 Report

Comments and Suggestions for Authors

The manuscript "Xanthone Rich Extracts from Two Gentiana Species: Promising Antidiabetic Potential" is well-written and concise. The hypothesis and purpose of the study are clearly and succinctly presented.

Minor mistakes to be corrected:

- The legend for Figure 5 is incomplete; the names of the extracts are missing.

Additional observations:

- The justification for selecting the four extracts for in vivo tests is not adequate. because the best results were observed for Gd6 (although the results are not significantly better than the extracts Gd4 and Gd5) and for Gu4.

- In Figure 8, are the values for the area under the curve at 60 minutes or 180 minutes? If they are for 60 minutes, this should be specified.

- How was the blood collected from the rats for the determination of blood sugar levels?

Comments on the Quality of English Language

Some minor English revisions are needed.

Author Response

Thank you very much for taking the time to review this manuscript. Please find the detailed responses below and the corresponding revisions/corrections in track changes in the re-submitted files. We believe consideration of the respective remarks has helped to complement and improve this manuscript.

Comments 1: The legend for Figure 5 is incomplete; the names of the extracts are missing.

Response 1: In the legend for Figure 5 it is already stated that the represented chromatograms originate from G. utriculosa extracts derived from transgenic shoots (Gu3) before (top chromatogram) and after reaction with DPPH radicals (bottom chromatogram).

Comments 2: The justification for selecting the four extracts for in vivo tests is not adequate. because the best results were observed for Gd6 (although the results are not significantly better than the extracts Gd4 and Gd5) and for Gu4.

Response 2: Thank you for pointing this out. When deciding which two extracts of G. dinarica to use for animal treatment, we chose between the three root extracts (Gd4, Gd5, Gd6) that were most successful in the DPPH assay and in alpha glucosidase inhibition. Because of the fact that the quantitative and qualitative composition of secondary metabolites changes depending on the way a certain part of the plant was obtained, we chose Gd4 and Gd5 because these two extracts were obtained from the same part of the plant (root) but with different growing techniques (Gd4-in vitro propagation and Gd5-genetic transformation). As for G. utriculosa, the reviewer correctly observed that the most successful Gd4 extract was not chosen for the treatment of animals. In this case, we wanted to be consistent in the selection of extracts for both plants. So we chose two extracts of G. utriculosa (Gd2 and Gd3) that were among the best and that met the criteria of coming from the same part of the plant but grown in two different ways (Gu2-in vitro propagation and Gu3-genetic transformation), as in the case of G. dinarica.

Accordingly, the necessary changes have been made in the revised manuscript (page 11, paragraph 2.4., highlighted in yellow).

Comments 3: In Figure 8, are the values for the area under the curve at 60 minutes or 180 minutes? If they are for 60 minutes, this should be specified.

Response 3: Thank you for pointing this out. The values for the area under the curve are for the first 60 min. Accordingly, the necessary changes have been made in the legend for Figure 8 (page 12, highlighted in yellow).

Comments 4: How was the blood collected from the rats for the determination of blood sugar levels?

Response 4: Blood sample were obtained from the rat tail vein. Less than 2 mm of tissue was cut from the tail tip, distal to the bone, with sharp scissors. Blood was obtained by direct flow or by gently massaging the tail and collecting the blood directly on a glucose test strip of blood glucometer (Accu-Chech Active, India).

This explanation has been incorporated into the revised manuscript (page 18, paragraph 4.6., highlighted in yellow).

Comments 5: Some minor English revisions are needed.

Response 5: We have corrected all language errors throughout the manuscript.

Reviewer 2 Report

Comments and Suggestions for Authors

1. The in vivo antidiabetic potential was only assessed through a basic phenotypic study, lacking further research into underlying mechanisms and other aspects.

2. How was the dosage of the extracts determined for the in vivo trials?

3. Why did the authors choose to use mean ± standard error instead of mean ± standard deviation

4. The title of this manuscript is too broad; it is recommended that the authors make further revisions.

Comments on the Quality of English Language

None

Author Response

Thank you very much for taking the time to review this manuscript. Please find the detailed responses below and the corresponding revisions/corrections in track changes in the re-submitted files. We believe consideration of the respective remarks has helped to complement and improve this manuscript.

Comments 1: The in vivo antidiabetic potential was only assessed through a basic phenotypic study, lacking further research into underlying mechanisms and other aspects.

Response 1: We started this study with the aim to investigate and compare antidiabetic properties of methanolic extracts prepared from different parts (aerial part or root) of Gentiana dinarica and Gentiana utriculosa, cultured in vitro and genetically transformed with those collected from the nature. The obtained results revealed that xanthone rich extracts have the most prominent antioxidant and antihyperglycemic effect. This study allowed us to narrow down the selection from the initial 10 extracts to only the 4 most successful ones. In this way, we laid a good foundation for the next study, in which we will deal with the mechanisms of antidiabetogenic action on the experimental model of diabetes in rats.

Comments 2: How was the dosage of the extracts determined for the in vivo trials?

Response 2: In our previous study (Hepatoprotective effects of Gentiana asclepiadea L. extracts against carbon tetrachloride induced liver injury in rats by Vladimir Mihailović et al., Food and Chemical Toxicology 52 (2013) 83–90) we orally administered methanol extracts of aerial parts (GAA) and roots (GAR) of G. asclepiadea at doses of 100, 200, and 400 mg/kg b.w. to Wistar rats in order to evaluate its hepatoprotective potential. Significant liver protection against the toxicant (carbon tetrachloride) was more pronounced in the rats treated with GAR at the highest dose (400 mg/kg b.w.). Bearing this in mind and bearing in mind the similarity in composition of extracts of Gentiana asclepiadea and Gentiana dinarica we conducted pilot experiment on oral glucose loaded rats with chosen extracts (Gu2, Gu3, Gd4, Gd5) in two different concentration (200 and 400mg/kg b.w.) on a smaller number of animals (n=3). This pilot experiment showed that a more effective dose for all selected extracts was 400 mg/kg b.w.

Decision to use the selected extracts in a dose of 400 mg/kg b.w. was also supported by a study of Ghazanfar et al. (Journal of Complementary and Integrative Medicine. 2017; 20170002; DOI: 10.1515/jcim-2017-0002). In this study the methanolic and hydroethanolic extracts (each at the dose level of 500 mg/kg b.w.) of Gentiana kurroo Royle were found to overcome the main symptoms of the diabetes. Similar results were observed in Gentiana olivieri the methanolic crude extract at a dose of (300 and 600 mg/kg body weight) administered orally for 28 days (Sezik E et al. Hypoglycaemic activity of Gentiana olivieri and isolation of the active constituent through bioassay-directed fractionation techniques . Life Sci. 2005;76:1223–12238).

Comments 3: Why did the authors choose to use mean ± standard error instead of mean ± standard deviation

Response 3: Standard deviation and standard error are commonly included in data analysis. While the actual calculations for standard deviation and standard error look very similar, they represent two very different, but complementary, measures. Standard deviation tells us about the shape of our distribution, how close the individual data values are from the mean value. The standard error is calculated by dividing the standard deviation by the sample size's square root. It gives the precision of a sample mean by including the sample-to-sample variability of the sample means and tells us how close our sample mean is to the true mean of the overall population. It is inversely proportional to the sample size; the larger the sample size, the smaller the standard error because the statistic will approach the actual value.

We used the standard error as a part of inferential statistics and a measure of how variable the mean will be, if we repeat the whole study many times.

Comments 4: The title of this manuscript is too broad; it is recommended that the authors make further revisions.

Response 4: We agre. The title has been changed to : „Evaluation of the antidiabetic potential of xanthone rich extracts from Gentiana dinarica and Gentiana utriculosa” (highlighted in yellow)

Comments on the Quality of English Language

None

Round 2

Reviewer 2 Report

Comments and Suggestions for Authors

The author has addressed my concerns. At the same time, it is suggested that the author can further improve the aesthetics of the charts in the article